# Eco-Efficiency as a Decision Support Tool to Compare Renewable Energy Systems

**Dominik Huber [1],\*** , **Ander Martinez Alonso [1,2]**, **Maeva Lavigne Philippot [1]** and **Maarten Messagie [1]**

1   Electric Vehicle and Energy Research Group (EVERGI), Mobility, Logistics and Automotive Technology Research Centre (MOBI), Department of Electrical Engineering and Energy Technology, Vrije Universiteit Brussel, Pleinlaan 2, 1050 Brussels, Belgium; ander.martinez.alonso@vub.be (A.M.A.); maeva.philippot@vub.be (M.L.P.); maarten.messagie@vub.be (M.M.)
2   Graduate School of Maritime Sciences, Kobe University, 5-1-1, Fukae-minami, Higashinada, Kobe 658-0022, Japan
\*   Correspondence: dominik.huber@vub.be; Tel.: +32-2629-3843

**Abstract:** Even though eco-efficiency (EE) is already applied to various energy systems, so far, no study investigates in detail the hourly, marginal and seasonal impacts of a decentralized energy system. This study assesses the hourly EE of the Research Park Zellik (RPZ), located in the Brussels metropolitan area for 2022 composed of photovoltaic installations, wind turbines and batteries. A cradle-to-grave life cycle assessment (LCA) to identify the carbon footprint (CF) and a levelized cost of electricity (LCOE) calculation is conducted. An existing design optimization framework is applied to the RPZ. Consumption data are obtained from smart meters of five consumers at the RPZ on a one-hour time resolution for 2022 and upscaled based on the annual consumption of the RPZ. As the EE is presented as the sum of the CF and the LCOE, a lower EE corresponds to an economically and environmentally preferable energy system. In a comparative framework, the developed method is applied to two different case studies, namely, (i) to an energy system in Vega de Valcerce in Spain and (ii) to an energy system in Bèli Bartoka in Poland. The average EE of the RPZ energy system in 2022 is 0.15 per kWh, while the average EE of the Polish and Spanish energy systems are 1.48 and 0.36 per kWh, respectively. When analyzing four selected weeks, both the LCOE and CF of the RPZ energy system are driven by the consumption of the Belgian electricity grid mix. In contrast, due to the very low LCOE and CF of the renewable energy sources, in particular wind turbines, the RPZ energy system's EE benefits and lies below the EE of the Belgium electricity grid mix.

**Keywords:** eco-efficiency; life cycle assessment; levelized cost of electricity; renewable energy system; photovoltaic installations; wind turbines; lithium-ion batteries; Belgium electricity grid

## 1. Introduction

To obtain the ambitious targets set by the European Commission to mitigate climate change, the introduction of renewable energy technologies is perceived as one objective to decarbonize the national electricity grid mix [1]. By default, the national electricity grid mixes are evaluated in climate change (CC) per average production of a given year [2], while such metrics might serve well as an overall system evaluation and for country comparison, it lacks providing its consumers with more precise information on the electricity supply at certain hours [2]. Fossil fuel or nuclear power plants help supply electricity continuously over a long time. In contrast, the marginal electricity grid mix at a given hour is particularly interesting for renewable energy carriers due to their intermittent nature. To better understand the advantages of an energy system containing a higher share of renewable energy carriers compared to an energy system mostly including fossil and nuclear power plants, it is therefore important to increase the temporal resolution from the current average electricity grid mix towards an hourly time step.

This analysis could contribute to increase the understanding of the electricity consumers at what time their electricity is generated by renewable energy technologies. Hence, the electricity consumers would be enabled to schedule their consumption accordingly. Specifically, this analysis can be of interest to large electricity consumers, such as the production industry or to electric vehicle (EV) owners. Thereby, the consumers are provided with a decision support to limit their CC impact and save electricity costs. Environmental impacts of hourly, marginal electricity grid mix have already been investigated in the literature.For example, Messagie et al. (2014) calculated the hourly carbon footprint (CF) of the Belgium electricity grid mix in 2011 [3]. More recently, Bastos et al. (2023) included, apart from CC, nine other impact categories for assessing the hourly electricity grid mix of Italy [4]. Furthermore, the hourly electricity mix of an energy system entailing a high share of renewable energy carriers is determined by the climate and weather conditions of its location. Consequently, the hourly electricity mix of such an energy system is characterized by changes in seasons and geographic conditions. The seasonal impact was investigated in a study by Kiss et al. (2020). They assessed the hourly CC impact between 2018 and 2020, taking into account different temporal aggregation levels, such as daily, weekly and monthly for Hungary [2].

When modeling the energy systems, one main optimization criterion of those models is cost. Potential investors and other stakeholders are interested in understanding the investment and operating costs they are facing when installing such a system. However, with the Paris Agreement aiming at limiting global temperature rise to well below 2 degrees above the pre-industrial level, much more emphasis is put on greenhouse gas emissions [5]. While the first energy system models (ESMs) focused on cost optimization, more and more studies are investigating the optimization of environmental impacts [6]. Instead of deciding between optimization of either costs or environmental impacts, the optimization function can be expanded to a multi-optimization problem covering both costs and environmental impacts, e.g., by conducting a life cycle assessment (LCA) [6]. Thereby, the level of integrating LCAs into ESMs can be distinguished. More and more research is dedicated to understand this integration process. For example, Astudillo et al. (2018) reviewed 10 studies combining bottom-up optimization ESMs and LCAs. They highlight the data mapping, the double-counting of energy demand and the lacking integration of life cycle impacts in the optimization problem, among others, as problematic [7]. Furthermore, Volkart et al. (2018) studied the integration of LCAs and ESMs and applied them to world energy scenarios. As one main observation, they stressed the negligence of technological advancements and the lack of data for all industrial and service sectors [8]. Recently, Blanco et al. (2020) examined LCA integration into an ESM and applied it to power-to-methane in the European Union [9]. The study identified the missing feedback loop of LCA results back to the ESM and the scarce technological details of industry data [9]. As mentioned, the integration of LCA and ESMs is still subject to various obstacles. A comprehensive overview of LCA studies conducted for different energy systems is included in Table A1 in the Appendix A.

Another alternative to avoid overlooking costs or environmental impacts is eco-efficiency (EE), representing a relationship between any economic and environmental variable, e.g., in the form of a ratio [10]. The EE was already applied for comparison of the different European electricity grid mixes. For example, Ewertrowska et al. (2016) integrated a data envelopment method and an input–output model [11]. Another study evaluated the EE of 28 EU countries, employing a combination of a data envelopment method and an LCA [12].

The studies highlight the importance of a finer temporal resolution when evaluating different electricity mixes, show LCA integration into ESM and the application of EE. At the same time, these studies are limited in their scope, e.g., they all focus on a national level. Hence, an evaluation of a specific energy system, such as an industrial park or a residential neighborhood, remains unaddressed.

As a consequence, this article investigates, to the authors' knowledge, for the first time the seasonal and geographical differences of an energy system in Belgium based on

the hourly EE. Therefore, electricity consumption data for 2022 are obtained from five smart meters and are up-scaled to represent the consumption of all consumers located at the Research Park Zellik (RPZ) in Belgium. Additionally, the life cycle inventory of the energy system is obtained from an internally developed framework, showcasing the interaction between an ESM and LCA. To evaluate the seasonal impact, the hourly EE of the RPZ is compiled for a spring, summer, autumn and winter week in 2022. Besides the seasonal impact on the EE, the performance of the RPZ energy system is also compared to the Belgium electricity grid mix and to two other energy systems located in Poland and Spain to evaluate the spatial difference of the EE.

Therefore, this study addresses the four following research questions:

1. How do electricity production, electricity consumption and the EE of the RPZ energy system vary during a winter, summer and transition week within 2022?
2. On an hourly basis, what are the main contributing assets to the EE of the RPZ energy system in 2022?
3. How does the RPZ energy system perform in terms of EE compared to consuming the current Belgium electricity grid mix in 2022?
4. What is the impact of different geographical locations on the EE?

## 2. Materials and Methods

The aim of this study is to compile the EE of a reference energy system. Thus, the following research design is proposed: In the first step, yearly data distribution is analyzed through the design and optimization framework, a Python-based virtual platform developed at the Vrije Universiteit Brussel (VUB). This framework allows the technical, economic and environmental evaluation of an energy system integrating assets of different natures [13]. Data called from this framework are used in an LCA to compile the carbon footprint (CF) and calculate the levelized cost of electricity (LCOE). Hereafter, the CF and the LCOE are combined to determine the EE for the reference energy system (see Figure 1). Furthermore, a comparative framework is developed, including a broad sensitivity analysis and three different energy system case studies. To understand the results' sensitivity towards certain parameters, a local sensitivity analysis in the form of a linear regression analysis is conducted. Furthermore, the EE is tested for a Polish and Spanish case study, including a comparison of the respective electricity grid mix.

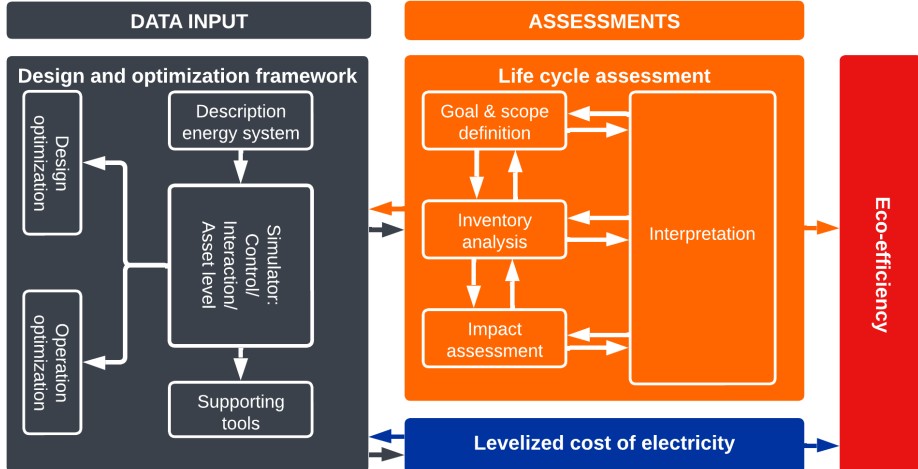

**Figure 1.** Research design for an assessment of an energy system. On the left, the functional design of the design and optimization framework is presented. This framework provides input for the subsequent assessments (middle rows). All evaluations are then used to compile a final eco-efficiency. Source: own compilation and [13].

*2.1. Design and Optimization Framework*

The energy system is sized and optimized by applying a VUB internally developed design and optimization framework. Its objective is to simulate and optimize the design and operation of an energy system, taking into account various electric assets and evaluating its technical, economic and environmental performance. Thereby, the center of the virtual platform is an energy system simulator, reproducing assets based on state-of-the-art models from the literature and particularly developed models. First, the framework determines optimal asset sizes by analyzing the power dispatch over a given time horizon. The objective of the optimization can be defined between different key performance indicators. For the present paper, the objective function is to minimize the total cost, comprising investment and operation costs. Once the minimization objective is selected, the problem is defined as a mixed integer linear problem. The methodology of the optimization problem formulation is presented comprehensively in Felice et al. (2022) and will therefore not be further detailed as part of this study [13].

The framework is composed of four main modules: (i) the config file, (ii) the simulator, (iii) design and operation optimization and (iv) supporting tools (see the left side of Figure 1). The config file describes the structure of the energy system and is used as an input file in the simulator. The simulator is built on three levels: (i) the control level taking the decisions for the remaining levels, (ii) the interaction level regulating the interaction and information exchange and (iii) the asset level, where assets are modeled. The output is then a design and operation optimization for the energy system. To support the simulator, weather data are imported, a calculation tool for tariffs and costs is added, a configuration file generator is set in place and the developments are reported.

*2.2. Levelized Cost of Electricity*

Based on the inputs of the design and optimization framework (DOF), the first step is to calculate the LCOE, represented as the ratio of the installation, operation and end-of-life (EoL) treatment costs over the cumulative lifetime discharged electricity (Equation (1)):

$$LCOE\frac{EUR}{kWh} \ = \ \frac{c_{ES}}{e_{ES}},\tag{1}$$

where:

- $c_{ES}$ are the lifetime costs of the energy system (EUR);
- $e_{ES}$ is the lifetime cumulative discharged electricity of the energy system (kWh).

In order to show the CF and LCOE variance over the year, four weeks are selected to represent the four seasons. The weeks containing the 20 March, the 21 June, the 21 September and the 21 December 2022 are chosen to represent spring, summer, autumn and winter, respectively. These weeks are analyzed in more detail in Section 4. First, the CF and capital expenditures (CAPEX) of the respective assets are calculated for the sizes determined by the DOF. Next, the total CF and CAPEX are divided by the cumulative amount of generated or stored electricity and multiplied by the electric quantity at each specific hour to obtain the CF and CAPEX per hour of the representative weeks in 2022. Third, the hourly values are then divided over the functional unit (FU), namely, the consumed electricity.

*2.3. Life Cycle Assessment*

First, to determine the CF of the energy system, the Intergovernmental Panel on Climate Change (IPCC) 2013 CC is selected as the life cycle impact assessment method [14]. The FU for the CF and the LCOE is one kilowatt hour (kWh) of consumed electricity in 2022 by the reference energy system. The function of the energy system is to provide electricity for over 25 years. An attributional LCA following a cradle-to-grave approach is chosen (see Figure A2 in the Appendix C, applied to the Belgian case study). Raw material extraction and asset manufacturing, the use stage and EoL treatment of the assets are included.

A combination of the Python package Brightway 2 and the activity browser, in combination with the ecoinvent database version 3.8 as a background database, is used [15–17]. Further details on life cycle inventories (LCIs) are provided in the Appendix B. This study does not aim to provide a functional system layout from a technical point of view. Hence, only relevant assets are included. Due to less significance of environmental impacts, smaller components, e.g., the balance of the system unit, cables, casing, etc., are ignored [18–20]. Additionally, it is estimated that the inverter can function as a hybrid inverter and not every single asset is equipped with its single inverter. Furthermore, the assessment focuses on the generated and stored electricity and its overall consumption. However, the study does not investigate which appliances are supplied with the generated and stored electricity. To that extent, EV charging or any other charging strategy, such as smart charging, is included in the assessment only in the form of the overall electricity used to charge the EVs at a certain time. Hence, neither the charging stations nor the EVs themselves are included in the assessment. Table 1 summarizes the goal and scope of this study.

**Table 1.** Summary of the goal and scope description (RPZ = Research Park Zellik; PV = photovoltaic; FU= functional unit; LCIs = life cycle inventories; LCIA = life cycle impact assessment; IPCC = Intergovernmental Panel on Climate Change; GWP = global warming potential; LCOE = levelized cost of electricity).

| Parameter | Description |
|---|---|
| Selected method | Attributional LCA |
| Product system | Electricity system of the RPZ, Brussels, comprising PV installations, wind turbines and batteries |
| The function of the product system | Providing the RPZ with electricity over the next 25 years |
| System boundaries | Cradle-to-grave |
| Functional unit (FU) | 1 kWh of consumed electricity |
| Life cycle inventories (LCIs) | Foreground system: PV installations: [17,21] Wind turbines: [17] Batteries: [22,23] Belgian grid: electricity map [24] Background system: ecoinvent 3.8 [17] |
| Life cycle impact assessment (LCIA) | IPCC 2013, Climate change, GWP 100 a |

### 2.4. Eco-Efficiency

According to the ISO 14045 standard, EE is defined as an indicator to measure the environmental impacts of one product system compared to the value of another product system [25]. Over the years, the EE calculation evolved, resulting in plenty of variations of these metrics [10]. A considerable research effort is dedicated to applying the EE in various contexts [12,26–28]. For this study, the EE is computed as the sum of LCOE and CF. Hence, the most beneficial energy system is the one which presents the minimal sum of both the CF and the LCOE. Thereby, the suggested EE does not represent the original definition of efficiency, which is input over output. Instead, the purpose of the EE is to provide a single metric to compare different energy systems, computed from the CF and the LCOE. This approach is not new anymore and is applied in Suh, Lee and Ha (2005) [10,29]. Heijungs (2022) describes this procedure, particularly relevant for customer or clients that aim to minimize both their environmental impact and their costs at the same time [10]. Therefore, the EE is computed for each hour over one year by applying Equation (2):

$$EE = \sum(CF, LCOE), \tag{2}$$

where:

- $EE$ is the eco-efficiency $((kgCO_2eq + \text{EUR})/\text{kWh})$;
- $CF$ is the carbon footprint $(kgCO_2eq/\text{kWh})$;
- $LCOE$ is the levelized cost of electricity $(\text{EUR}/\text{kWh})$.

*2.5. Comparative Framework*

2.5.1. Case Studies

In order to study the performance of energy systems of different natures and locations, an analysis of three suitable case studies is performed: (i) The RPZ in Belgium, introduced as the reference energy system. It is a $CO_2$-neutral joint project of the VUB and the University Hospital of Brussels, represented by an industrial research park composed of more than 70 companies. The energy system constitutes the reference case study of the present study and comprises two dispatchable assets (PV and wind turbines) and two stationary battery storages, and it is linked to the Belgian electricity grid mix. (ii) Vega de Valcarce, a rural town in Spain, was studied in the context of the Renaissance project towards the implementation of an Energy Community [30]. At this location, the Spanish electricity grid mix and the integration of PV panels are addressed to supply a load, including the town hall and the school, as well as 100 residential and 2 commerce consumers. (iii) Bèli Bartoka, in Poland, is a residential complex comprising 128 apartments and 4 commerce consumers. It is an innovative project undergoing a continuous modernization process towards reducing its CF and energy consumption. The Polish electricity grid mix, PV and a wind turbine are considered for this energy system. The main assumptions of the three case studies are presented in Table 2. The presented case studies vary both in time and technical system layout. Similar to the RPZ energy system for the year 2022, four weeks are also selected for Vega de Valcarce and Bèli Bartoka, containing the date of the change of seasons. Results are explained in detail for the RPZ energy system. As the two other case studies are only selected for comparative reasons, they are not analyzed in detail. Hence, only the mean values of the EE per week of the energy system and the business as usual (BAU), namely, the consumption of the national electricity grid mix per country, are included.

**Table 2.** Technical overview of the three different case studies: BE—Research Park Zellik, ES—Vega de Valcarce, PL—Bèli Bartoka. Data input for the modeling of all case studies is provided on an hourly time resolution. However, to provide a clear overview, the values indicate the average values of the year-long time series. (PV = photovoltaic panels; source: own compilation and [30]).

| Site | Type | Assets | Grid Price (cEUR/kWh) | Grid Carbon Intensity (g $CO_2$/kWh) | Total Consumption (kWh/year) |
|---|---|---|---|---|---|
| ES | Public administration and residential | Grid and PV | 14.36 | 282 | 443,116 |
| PL | Residential and commercial | Grid, PV and Wind Turbine | 13.1 | 855 | 953,096 |
| BE | Industrial and office buildings | Grid, PV, Wind Turbine and Battery | 9.82 | 176 | 3,412,058 |

2.5.2. Sensitivity Analysis

To understand the influence of particular modeling parameters on the EE, a sensitivity analysis is performed. Therefore, specific parameters are selected and modified to understand their impact on the EE. A total of 11 critical modeling parameters are investigated: (i) the ES lifetime, (ii–iii) the energy density of the two batteries and (iv–xi) both CF and CAPEX of PV installations, wind turbines and two lithium-ion batteries (LIB). In particular, one LIB has a cathode made of lithium iron phosphate (LFP) and another LIB with a cathode

manufactured of lithium nickel manganese cobalt oxide (NMC). Out of a range from ±50% of the 11 selected parameters, 20 values are chosen randomly. Subsequently, their influence on the results is presented. In a final step, a linear regression analysis is conducted, which presents the correlation of a given modeling parameter with the EE.

*2.6. Data*

2.6.1. Manufacturing

To calculate the CF of the energy system, the capacities provided by the framework are linked with environmental inventory data from ecoinvent 3.8 [17] or LCI from the literature [22]. More specifically, datasets of 3 kWp single-Si PV installation, 2 MW wind turbine, one LFP battery and one NMC battery are selected. To convert the stationary battery storage capacity outputted by the DOF into a mass quantity, an energy density at pack level of 125 Wh/kg for LFP batteries and of 143 Wh/kg for NMC batteries is applied [31,32]. Due to the fact that the design and operation of the RPZ are already minimized towards costs, only a simple approach to calculate the LCOE is covered in this study. The lifetime costs of the energy system cover the initial CAPEX of the different assets (see Table 3). Those values are in line with the values chosen for the simulation with the DOF. For the Belgian electricity grid mix, no CAPEX costs are applied, as such costs that correspond to distribution and network infrastructure or the facility for producing electricity.

**Table 3.** CAPEX of the applied assets in the RPZ (CAPEX = capital expenditure; PV = photovoltaic; LFP = lithium iron phosphate; NMC = lithium nickel manganese cobalt oxide).

| Assets | CAPEX | Source |
|---|---|---|
| PV installation | 1000 EUR/kWp | [13] |
| Wind turbine | 1460 EUR/kW | [33] |
| LFP battery | 433.00 EUR/kWh | [34] |
| NMC battery | 443.00 EUR/kWh | [34] |
| Belgian grid infrastructure | 0.00 EUR/kWh | |

2.6.2. Use Stage

In this study, the energy system installed is assumed to last at least 25 years, while the dispatchable assets have similar or longer calendaring lives, the stationary battery storage most likely will require earlier replacement (see Table A3 in the Appendix B). The lifetime and the quantities of generated, stored and consumed electricity obtained from the framework are used to scale the manufacturing impacts to the FU. Further information on the use stage is provided in the Appendix B. The use stage costs only include the electricity costs of the Belgian, Polish and Spanish grids for the different cases. The day-ahead-forecast price for Belgium, Poland and Spain are derived for the respective years [35]. Those prices are selected, even though they do not cover distribution, transmission fees and taxes. Other costs and impacts, such as remuneration for injecting electricity back into the grid, are neglected.

2.6.3. End-of-Life Treatment

Life cycle inventories of Frischknect et al. (2020) are applied, accounting for the material and energy requirements to recycle PV installations [21]. For wind turbine treatment, environmental impacts are accounted for in terms of using recycled material in the manufacturing process [36]. Both battery cells of the LIBs are treated in a pyrometallurgical recycling process, accompanied by hydrometallurgical treatment of the slag, allowing recovery of 95% of copper, iron and nickel and 70% of lithium [23,37]. Further information on technical assumptions of EoL treatment is provided in the Appendix B. Contrary to the LCA, the LCOE calculation takes only initial investment costs and electricity costs from the Belgian grid into account. EoL treatment costs are neglected in this study due to the high uncertainty linked to future material prices. A good example to show the volatility of

the prices for materials is lithium carbonate; over the last 10 years, its price has multiplied by more than 3 [38]. Besides the LCIs mentioned in this section and the Appendix B, BE-specific LCIs are applied from Huber et al. (2023) [39]. The source code for all method compilations is provided in the Supplementary Materials.

## 3. Results

### 3.1. Validation and DOF Output

Figure 2 visualizes the aggregated consumption profiles of different entities at the RPZ of four weeks and the average consumption during the four quarters of 2022. The consumption is presented per quarter, and for the same resolution and time period, minimum and maximum hourly values are calculated. A visual validation check shows, on an hourly basis, how the selected week represents the other weeks of the year. The selected weeks are depicted in Figure 2 in red. During peak consumption, the consumption in the selected weeks is most of the days higher than the minimum week. In the second and fourth quarters of 2022, the consumption of the selected week approximates the maximum consumption, while in the third week, the consumption is close to the minimum consumption. Consequently, Figure 2 shows that the selected week presents both minimum and maximum consumption compared to other weeks of the season. Thus, the selected four weeks are assumed to be a valid representation of other weeks in the same year.

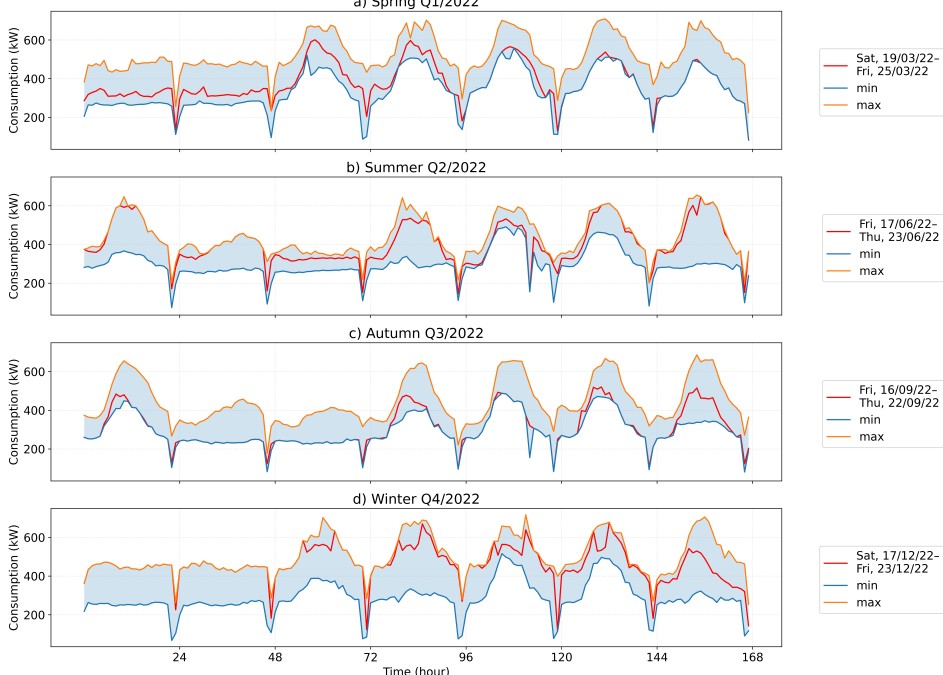

**Figure 2.** Consumption of the Research Park Zellik for four weeks in 2022 in (**a**) spring, (**b**) summer, (**c**) autumn and (**d**) winter.

Furthermore, based on cost minimization, the DOF resulted in the following asset sizes: 765 kWp PV installations and 690 kW wind turbines. Additionally, 2 stationary batteries, each with a capacity of 342.5 kWh, are included in the optimization model, as those are installed for demonstration purposes. Generated, stored and grid-consumed electricity at the RPZ is visualized in Figure 3 for 2022. In 2022, the main electricity source for the RPZ is the wind turbines, followed by the Belgian electricity grid mix and the PV installations. Both batteries provide significantly less electricity than the dispatchable assets and the electricity grid. As the electricity generation and storage are computed by the DOF, which follows an economic optimization, high wind-generated electricity is the most cost-efficient investment, as it produces electricity at low investment costs. In contrast, the DOF hardly uses stationary batteries. Therefore, their purpose can be questioned as

long as the energy system is still able to consume cheaper electricity from the national grid. However, stationary batteries might be fully valid in an isolated energy system without any consumption from the national grid. To satisfy the electricity demand of RPZ, smaller stationary batteries would have been enough. Thereby, an overall decrease in the system costs could be achieved. In the succeeding section, the seasonal impacts on electricity production and consumption and the interlinked CF and LCOE are explored in order to understand the computed EE over one year.

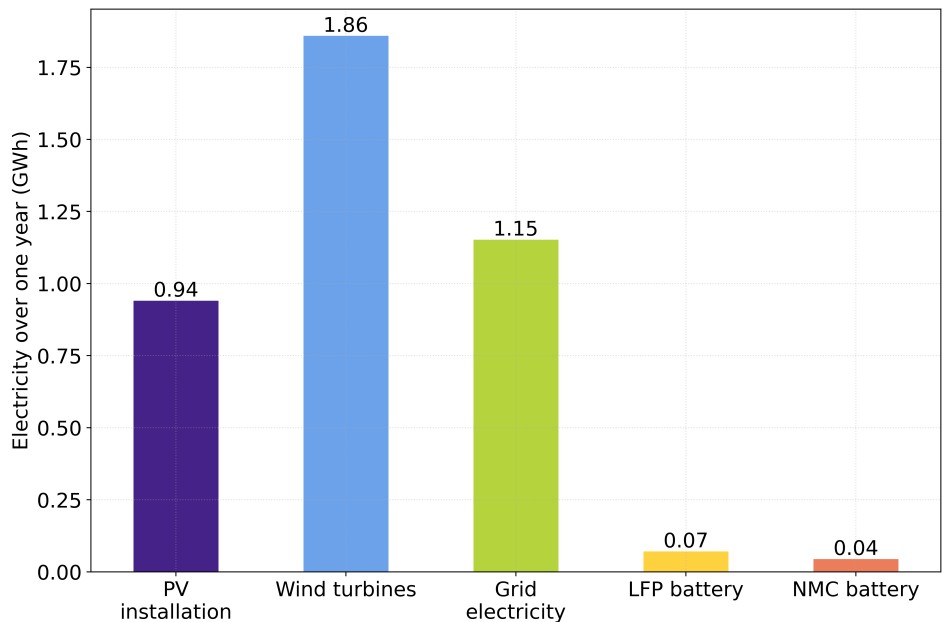

**Figure 3.** Generated and stored electricity over 2022 at the RPZ (PV = photovoltaic; LFP = lithium iron phosphate; NMC = lithium nickel manganese cobalt oxide; source: own compilation).

### 3.2. Seasonal Impacts

The selected week in spring is characterized by a weather change; from Monday to Wednesday, there is high wind production, almost enough to supply the RPZ demand and some PV electricity production during midday. During this time, no electricity is supplied by the Belgium electricity grid mix. Instead, the excess electricity from the wind turbine is fed back to the grid. From Thursday until the end of the week, wind electricity production declines, while PV electricity production grows during midday. However, PV-produced electricity is not sufficient to meet the RPZ demand every single hour. Thus, the remaining electricity is consumed from the Belgium electricity grid mix. At the same time, the batteries are charged and discharged to support the electricity supply. Following the weather conditions, at the beginning of the week, the wind turbines drive the CF of the RPZ energy system, supplemented by the CF of PV installations during midday. On a kWh basis, the wind turbines contribute the least to the CF. Consequently, the CF of the RPZ energy system outperforms the CF of the BAU. From Friday onwards, the CF of the RPZ approximates the BAU due to the higher CF of the Belgium electricity grid mix and in combination with the CF of the PV installations. Chart (b) in Figure 4 also shows that the renewable energy system at RPZ has, during most hours, a lower CF than the BAU. In terms of LCOE, similar observations compared to the CF can be made. In the absence of renewable energy source (RES) electricity production, the Belgium electricity grid mix accounts for the majority of the costs of the RPZ energy system. The LCOE of the RPZ remains in the spring week below the LCOE of the BAU, whereas the BAU performs worst in the selected week. These observations translate into the EE; at the beginning of the spring week, the EE of the RPZ energy system performs better than the EE of the BAU.

Towards the end of the spring week, the EE of the RPZ approximates the EE of the RPZ (see Figure 4).

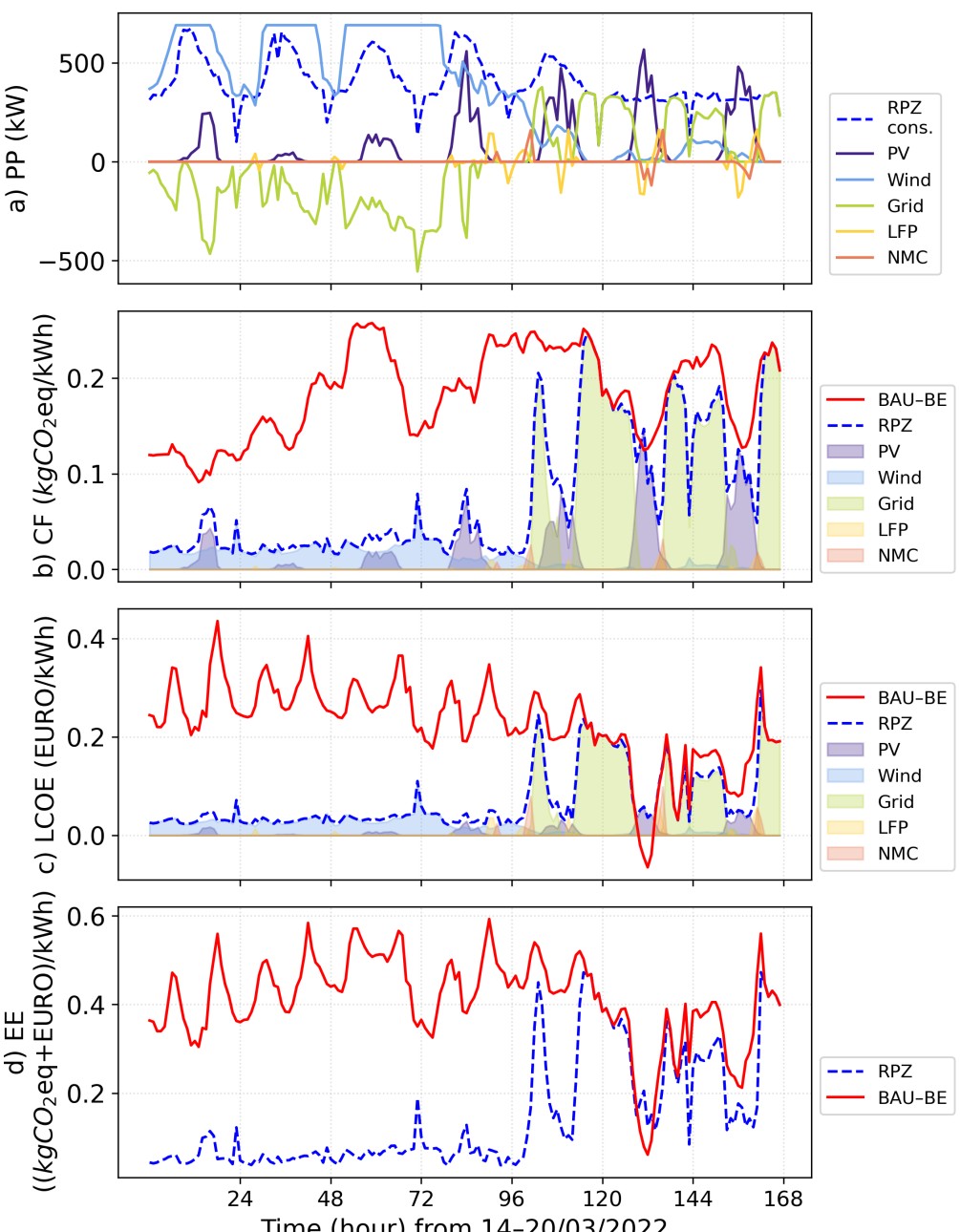

**Figure 4.** Hourly electric power production and consumption by the different assets of the RPZ (positive) and discharged electricity from the stationary batteries and electricity feed back to the grid (negative, subplot (**a**)), the CF (subplot (**b**)), the LCOE (subplot (**c**)) and the EE (subplot (**d**)) of a week in spring in 2022 (PP = power production; CF = carbon footprint; LCOE = levelized cost of electricity; EE = eco-efficiency; BAU = business as usual; RPZ = Research Park Zellik; cons. = consumption; PV = photovoltaic; LFP = lithium iron phosphate battery; NMC = lithium nickel manganese cobalt oxide battery; source: [13] and own compilation).

The selected summer week is characterized by high PV electricity production during midday and much lower but constant wind-generated electricity. Additionally, the intermittent nature of RES electricity generation requires the integration of battery storage to overcome the gap between production and consumption. Due to longer sunshine hours

per day, the peak production of PV installations is extended, while during off-peak production hours, electricity is supplied by the Belgium electricity grid mix. Only on Sunday, when consumption is lower compared to weekdays, is electricity fed back into the grid. The production of renewable electricity results in a considerable difference between the CF of the BAU and the RPZ energy system. Most of the CF peaks are due to the Belgium electricity grid mix, only exceeded twice by the CF of PV installations during PV's peak electricity production. Hence, the main contributors to the CF of the RPZ energy system remain the Belgium electricity grid mix, followed by the PV installations. Compared to PV installations and the Belgium electricity grid mix, wind turbines and stationary batteries account for a much lower share of the CF. Similar observations regarding the CF can be made for the LCOE during this summer week; the main cost driver is the LCOE of the Belgium electricity grid mix. In contrast, the other RES asset's contribution to the RPZs' LCOE is significantly lower compared to the Belgium electricity grid mix. Apart from the outbreaks of LCOE due to the Belgium electricity grid mix, the LCOE of the RPZ energy system is below the LCOE of the BAU. Generally, the EE of the BAU is, almost at every hour, higher compared to the EE of the RPZ (see Figure A2 in the Appendix C).

The autumn week represents a great mixture of electricity generation from different sources; the beginning of the week is defined by moderate PV electricity and low wind electricity production. As a result, significant electricity is supplied by the Belgium electricity grid mix during off-peak hours and supported by battery storage. During Tuesday, Wednesday and Thursday, both PV and wind electricity production increase, leading to lower consumption of Belgium's electricity grid mix. At the same time, battery storage is used, and the excess electricity during midday is fed back to the grid. On Saturday and Sunday, the wind-generated electricity is enough to satisfy the electricity demand of the RPZ. In fact, the combination of peak production of the PV installations and the wind-generated electricity results in a feed back of the excess electricity production into the Belgian grid. Compared to wind-generated electricity, stationary batteries are cycled on a much lower power level. At the beginning of the selected autumn week, the CF of the RPZ energy system is driven by the Belgium electricity grid mix and the PV installations. From Wednesday onwards, the CF of the RPZ energy system benefits from the uptake of wind-generated electricity in combination with a decrease in Belgium's electricity grid mix. On the weekend, the CF of the RPZ energy system is lowest due to the marginal CF contribution of wind-generated electricity. Similar observations for LCOE as for CF can be made; the LCOE of the RPZ energy system is below the LCOE of the BAU, only interrupted by the LCOE of the Belgium electricity grid mix during some off-peak hours at the beginning of the week. The magnitude of the LCOE of the RES assets is below the LCOE of the Belgium electricity grid mix. To the same extent, the EE of the RPZ remains below the EE of the BAU (see Figure A3 in the Appendix C).

In the winter week, wind-generated electricity production supplies most of the RPZ's electricity demand, supported by stationary battery installations. Additionally, PV-generated electricity is negligible due to low solar irradiation in winter. Compared to the production of wind-generated electricity, the utilization of stationary batteries occurs again, only at much lower power levels. At times of excess wind electricity production, it will be fed back into the Belgian grid. Generally, the CF of the RPZ energy system is significantly lower than the CF of the BAU. During the few hours when wind electricity production does not meet the RPZ demand, the Belgium electricity grid mix determines the CF of the RPZ energy system. This represents the steepest CF in this winter week. For the remaining time steps of the week, the CF of the RPZ energy system remains modest, thanks to the decrease in peak electricity production of the PV installations and the small CF of the other RES technologies. The few peaks of LCOE of the RPZ energy system are driven mainly by the costs for Belgium's electricity grid mix at times when the produced wind electricity is not sufficient to supply the RPZ demand. Besides those mismatches, the LCOE of the RPZ energy system is marginal due to the low LCOE of the wind turbines.

Moreover, the EE of the BAU remains higher than the EE of the RPZ during the entire week (Figure A4).

### 3.3. Geographical Impacts

As demonstrated in the previous section, the national electricity grid mix is an important factor to determine the EE of the case studies. The annual average CF of the national grid is 167, 855 and 283 g $CO_2$/kWh for Belgium (2022), Poland (2021) and Spain (2019). The LCOE of the national grids is 0.24, 0.09 and 0.05 EUR/kWh for Belgium (2022), Poland (2021) and Spain (2019). The EE of BB is driven by the high CF of the Polish electricity grid mix. The already high CF of the national electricity grid mix adds to the CF of the PV installations. Thereby, the highest EE during summer is computed. In winter, when no PV-generated electricity is included, the EE of the BB energy system equals the EE of the BAU-PL. In contrast to BB, the Spanish case study benefits from a lower CF and LCOE, making the Spanish electricity grid mix the most competitive of the three countries. As the annual PV electricity production at VV is only 881 kWh compared to an annual electricity consumption of 433,116 kWh, the CF and LCOE of the PV installation hardly contribute to the EE. Considering the Belgium case study, the EE of the BAU-BE is slightly inferior compared to the EE of BAU-ES, but considerably better than the BAU-PL. Even though the EE of the BAU-ES is lower compared to the BAU-BE, the RPZ energy system reveals the lowest EE of all analyzed systems (see Figure 5). The reasons are as follows: (i) The electricity-generating assets are sized large enough to cover the peak productions. (ii) There is a great overlap of peak production and peak consumption, hence the peak production can be supplied by RES electricity most of the time. (iii) The mix of RES technologies ensures the RPZ to take advantage of RES electricity production throughout all different times of the year. (iv) Finally, the high impact of Belgium electricity grid mix is only consumed to bridge off-peak consumption, hence the RPZ energy system benefits at that time from cheaper off-peak prices. At BB, which represents a residential site, there is only a very small overlap between peak PV electricity production and peak consumption, as it appears during off-peak PV electricity production. In summary, considering geographical differences reveals that the weather conditions might influence the RES production. Nevertheless, more important is the appropriate size of the involved assets, so they can support limiting the consumption of the national electricity grid mix. The EE of the RPZ energy system proves to be more beneficial than the BAU-BE, if enough RES-generated electricity is supplied during peak consumption. As seen in the seasonal analysis, the RPZ energy system ends up with a lower EE, because different technologies are in place to absorb different weather conditions. Hence the energy system benefits from low-investment assets, such as PV installations or wind turbines.

### 3.4. Sensitivity Analysis

The values on the x-axis in Figure 6 represent the EE of the RPZ while modifying one parameter at a time. The energy system lifetime does not only affect the electricity production and consumption but also takes into account the individual lifetimes of the assets. One first finding is that there is a positive, linear correlation between the EE and the estimated lifetime of the energy system. This correlation corresponds to a $R^2$ value of 0.999793 (see subplot (a) of Figure 6). Additionally, changing the lifetime results in changes in the EE between $-16.63\%$ and $+16.95\%$ of the mean EE for the RPZ. The energy densities of the LIBs are modified as second parameters, which impacts the amount of storage required to store 1 kWh of electricity. Thereby, a higher energy density would result in a lower amount of needed LIBs and vice versa. As visible in subplot (b) of Figure 6, the modification of the energy densities has limited influence on the EE. Variations are found to result in a range of EE of $-0.61\%$ up to $+1.11\%$ of the mean EE for the RPZ. Due to the fact that the LFP battery is used more than the NMC battery, subplot (b) of Figure 6 illustrates lower EE modification for the NMC than for the LFP batteries. Additionally, the negative correlation between the energy density and the EE is presented,

corresponding to $R^2$ values for the density of NMC and LFP batteries of 0.893198 and 0.809902, respectively. As the last two parameters, the sensitivity of the EE towards the CF and the CAPEX of the assets is evaluated. Both parameters demonstrate a positive, linear correlation. Particularly high $R^2$ values are observed for the CAPEX of all assets. Here, the $R^2$ values vary from 0.999062, 0.997248, 0.981881 and 0.996207 for the PV installations, the wind turbines and the LFP and NMC batteries, respectively. However, the impact on the EE of changing the CF and CAPEX differs when considering the different assets. Similar to the energy density, the impact on the EE of different CF and CAPEX for the LIBs is limited, compared to the PV installations and wind turbines. Modifying the CF and the CAPEX of the PV installations has the greatest impact on the EE. Compared to the PV installations, the impact of the wind turbines on the EE is smaller, especially when varying the CF. Altogether, the sensitivity analysis proved the directly proportional relationship between the most important parameters and the calculated EE. Hence this model can be regarded as a linear model. Furthermore, the energy system's lifetime is found to have the strongest influence on the EE.

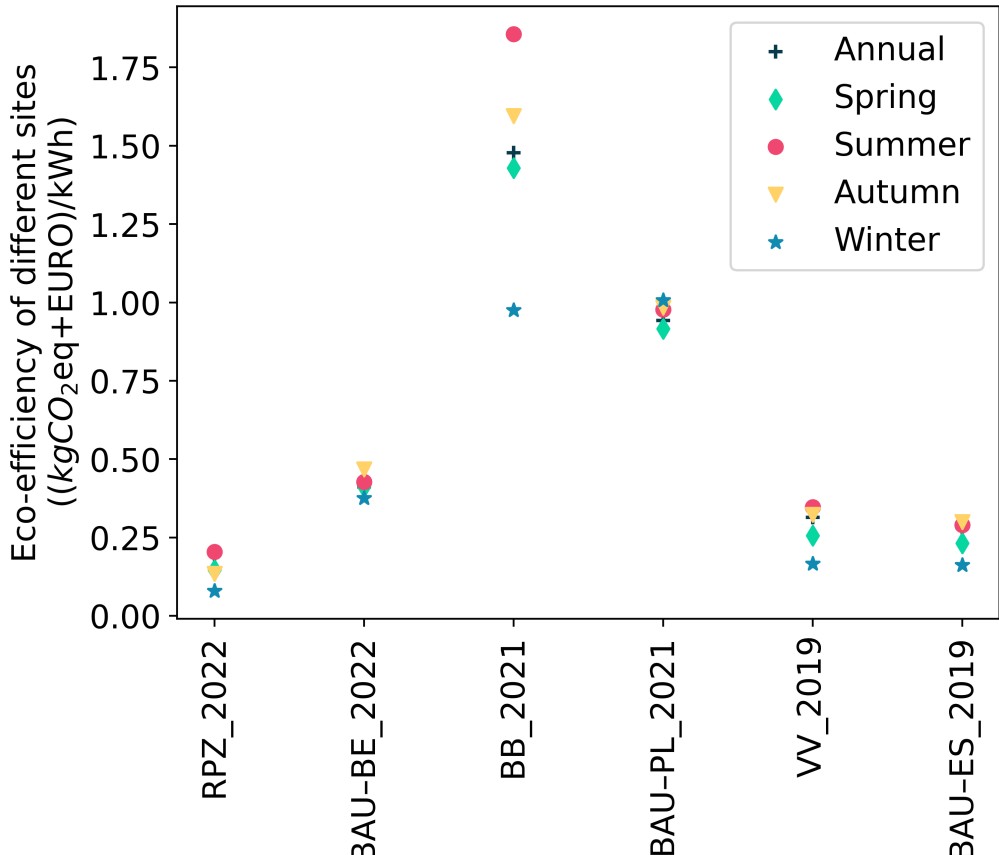

**Figure 5.** Eco-efficiency of three different energy systems and three different national electricity grid mixes (RPZ = Research Park Zellik; BE = Belgium; BB = Bèli Bartoka, the Polish residential site; PL = Poland; VV = Vega de Valcerce, the Spanish residential site; ES = Spain).

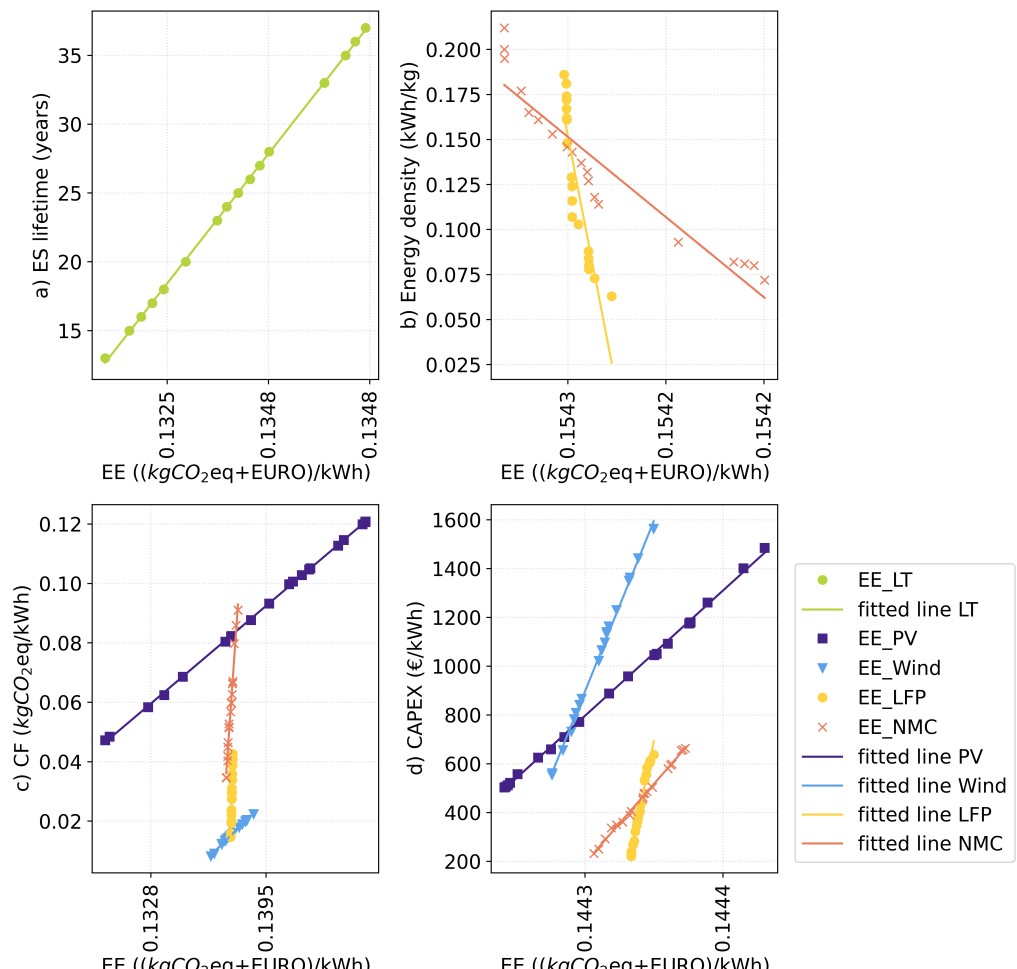

**Figure 6.** Sensitivity analysis of the computed eco-efficiency calculation for the RPZ. (LT = lifetime; ES = energy system; EE = eco-efficiency; PV = photovoltaic; LFP = lithium iron phosphate; NMC = lithium nickel manganese cobalt oxide; CF = carbon footprint; CAPEX = capital expenditure).

## 4. Discussion

This study is subject to some temporal limitations. Those include the different time horizons of the LCA and the DOF, the neglected asset degradation and their technological advancements and the comparison of the different energy systems from different years. Another point to mention is the upscaling of assets for the LCA. For this study, linear upscaling of all assets based on the capacity is applied. In fact, the sizing of the assets might not always be linear, and the number of materials might diverge. In addition, the LCOE calculation is very valid for comparison reasons. However, in practice, such great investments might not always be feasible, as it is required before commissioning the energy system. Thus, even though the optimally sized energy system is proven to be cost-advantageous, the initial investment might exclude disadvantaged groups from renewing their energy system. Moreover, the tariff structure of the national electricity grid mixes is simplified for this study. First, instead of day–night tariffs, which are commonly contracted for energy consumers, day-ahead prices are utilized. However, only very large entities or balanced responsible parties are interacting in this market, while the final energy consumers are offered day–night tariffs. Second, the day-ahead prices do not represent the final prices of the consumers. The final prices will be supplemented by costs for distribution, transmission and taxes. Third, no remuneration for feeding back electricity to the Belgian grid is considered.

Nevertheless, this study finds the EE as a suitable assessment for economic and environmental comparison of small-scale energy systems, such as the RPZ, VV and BB.

The calculation becomes particularly relevant for the end consumer of the energy system. By identifying the EE on a one-hour time resolution, the consumers are provided with more comprehensive information about the economic and ecologic performance of their energy system at each specific point of time. Based on such information, the consumers can schedule their consumption towards times when the system provides electricity at a low EE, resulting in both decreased electricity costs and CF. Besides the consumers, this study is also relevant for future research, e.g., it paves the way towards a more thorough integration of LCA and ESM. Additionally, the study highlights two important points: First, the benefits of high RES technology integration, especially at the RPZ are shown. This case study stands out due to its technology mix, which supports RES electricity supply in times of peak consumption. Contrary to the expectation, that the energy system at VV would present the lowest EE due to particularly high solar irradiation, the RPZ energy system ends up with a lower EE. This underlines the importance of a broad RES technology mix and of appropriate asset sizes. Second, the study also points out the current national electricity grid mixes as main driver for the EE. However, with the ongoing decarbonization of the EU energy system, the impacts of the national electricity grid mixes are expected to decline within the next decades. In fact, a future decarbonization of the included electricity grid mixes could result in an improved EE. Hence the consumption of electricity grid mix could be made more attractive for their consumer, both financially and environmentally. How the future decarbonization of the national electricity grid mixes influence the EE and which role energy systems as evaluated in this study play, remains subject to future work.

## 5. Conclusions

This study presents the compilation of the EE of the RPZ in Brussels, Belgium and compares it to electricity supplied by the Belgian electricity grid mix. In the first step, a VUB internally developed DOF is applied to the RPZ in order to compile the hourly electricity production in 2022 of different assets. To generate electricity, PV installations and wind turbines are chosen, supported by two LIBs and consumption of the Belgian electricity grid mix. The RPZ load profile is obtained and scaled up from the smart meters of five RPZ consumers. Second, a cradle-to-grave, attributional LCA of this energy system is conducted and presents hourly CF per kWh of consumed electricity. Third, the CAPEX costs of each asset are used to determine the hourly LCOE of the same system. Fourth, the final EE per hour is determined. Moreover, four weeks are analyzed in more detail to understand the seasonal impacts on electricity production and consumption and the EE. Fifth, the calculated EE for RPZ is compared to two other case studies: (i) Vega de Valcarce in Spain and (ii) Bèli Bartoka in Poland.

The results indicate that most of the electricity is produced by the wind turbines, followed by the PV installations and the Belgian electricity grid mix. In contrast, the two LIBs provide significantly less electricity to the RPZ consumers. Analyzing the hourly CF and CAPEX of the spring, summer, autumn and winter weeks revealed the consumption of the Belgium electricity grid mix as the main contributor to the EE. At the same time, the EE benefits from the low impacts of the RES technologies. In 2022, the average EE of the RPZ is 0.15 per kWh, whereas the average EEs of BB and VV are 1.48 and 0.36 per kWh. Exploring further the selected weeks shows that the RPZ energy system performs considerably better compared to the BAU, especially during hours when wind turbines generate electricity. Overall, the EE of the RPZ energy system in the selected weeks is almost always below the EE of the BAU.

Moreover, the results prove the following: First, the stationary batteries are oversized for their current application, as they are never fully charged. In 2022, the maximum charge of the LFP and the NMC batteries is 162.69 kW and 237.52 kW at a given hour, while the capacity of each battery is 342.5 kWh. Either the capacity is adjusted to the actual need, or the unused capacity can be used for other services, such as participation in the imbalance market. For both options, the RPZ consumers would be provided with a monetary benefit, either in the form of decreased investment cost due to smaller capacities or by bringing

them additional earnings. Second, among the four weeks, the EE is lowest in the winter week. Notably, the EE of the RPZ benefits from low CF and LCOE, compared to other RES technologies or the Belgium electricity grid mix. Third, the EE of the RPZ energy system surpassed the EE of BAU during most hours. In fact, the analysis shows that the EE of the RPZ is below the EE of BAU; thus, the RPZ consumer can take advantage of these insights and benefit, both financially and environmentally. Fourth, the assessment of different sites revealed that the energy system benefits from various appropriately sized assets, and hence a mix of various technologies. This will allow the energy system to absorb different weather conditions. Fifth, a local sensitivity analysis proved that there is a correlation between the energy systems lifetime, the energy density, the CF and the CAPEX of each asset and the EE. The energy systems lifetime is revealed as the most sensitive parameter.

Nevertheless, this study is subject to certain limitations: (i) no asset degradation is included, nor (ii) technological advancement, or (iii) long-term weather forecasting is investigated, (iv) different data are chosen to simulate the case studies, (v) the LCIs of the different assets are linearly upscaled, (vi) no compensation for injecting electricity back to the Belgian electricity grid is considered and (vii) customized electricity tariffs for the RPZ consumers are neglected. Next, investigations can focus on further methodological advancements. For example, those can be the alignment of the LCI used in the LCA and the data used in the DOF, the creation of a loop of the environmental impacts back to the optimization model or even the extension of the cost-optimization model towards a multi-objective one. Furthermore, both system boundaries and the geographical scope of the study can be extended to further investigate the role of the entire energy system and see the impact of different locations.

**Supplementary Materials:** Applied calculations including the source code are made available at https://doi.org/10.5281/zenodo.7947791 (accessed on 10 May 2023).

**Author Contributions:** Conceptualization, D.H., M.L.P. and M.M.; methodology, D.H., M.L.P. and M.M.; validation, D.H., A.M.A. and M.L.P.; formal analysis, D.H.; investigation, D.H. and M.L.P.; resources, M.M.; data curation, D.H. and A.M.A.; writing—original draft preparation, D.H. and A.M.A.; writing—review and editing, D.H., A.M.A., M.L.P. and M.M.; visualization, D.H., A.M.A. and M.L.P.; supervision, M.L.P. and M.M.; project administration, M.M.; funding acquisition, M.M. All authors have read and agreed to the published version of the manuscript.

**Funding:** This research was funded by Vlaams Agentschap Innoveren & Ondernemen (VLAIO) in the framework of the MAMuET and OPTIMESH project.

**Conflicts of Interest:** The authors declare no conflict of interest. The funders had no role in the design of the study; in the collection, analyses, or interpretation of data; in the writing of the manuscript; or in the decision to publish the results.

## Abbreviations

The following abbreviations are used in this manuscript:

| | |
|---|---|
| *BAU* | Business as usual |
| *BB* | Bèli Bartoka |
| *CAPEX* | Capital expenditures |
| *CC* | Climate change |
| *CF* | Carbon footprint |
| *DOF* | Design and optimization framework |
| *EE* | Eco-efficiency |
| *ES* | Energy system |
| *ESM* | Energy system model |
| *EoL* | End-of-life |
| *EV* | Electric vehicle |
| *FEU* | Freshwater eutrophication |
| *FU* | Functional unit |

| | |
|---|---|
| *GWP* | Global warming potential |
| *IPCC* | Intergovernmental Panel on Climate Change |
| *LCA* | Life cycle assessment |
| *LCI* | Life cycle inventories |
| *LCIA* | Life cycle impact assessment |
| *LCOE* | Levelized cost of electricity |
| *LFP* | Lithium-ion battery with lithium iron phosphate as cathode material |
| *LIB* | Lithium-ion battery |
| *LT* | Lifetime |
| *NMC* | Lithium-ion battery with lithium nickel manganese cobalt oxide as cathode material |
| *PP* | Power production |
| *PV* | Photovoltaic |
| *RES* | Renewable energy source |
| *RPZ* | Research Park Zellik |
| *VUB* | Vrije Universiteit Brussel |
| *VV* | Vega de Valcerce |

## Appendix A. Literature

**Table A1.** Overview and summary of most relevant studies (CF = characterization factor; FU = functional unit; IC = impact categories FE = freshwater eutrophication; HTP = human toxicity potential; PM = particulate matter; PO = photochemical oxidants; TA = terrestrial acidification; TE = terrestrial ecotoxicity; PV = photovoltaic; EPBT = energy payback time; NER = net energy ratio; CEF = carbon emission factor; LCIA = life cycle impact assessment; LCOE = levelized cost of electricity; GWP = global warming potential; US = United States of America; ESM = energy system model; source: own compilation).

| Reference | Analyzed System | Opportunities and Limitations |
|:---:|:---|:---|
| [40] | • Hybrid optimization of multiple energy resources (HOMER)<br>• Kenya, Africa<br>• 3 microgrids: (i) PV-battery, (ii) PV-generator, (iii) PV-hybrid<br>• FU: 1 kWh of electricity consumed<br>• Cradle-to-grave<br>• IC: CC, FE, HTP, PM, PO, TA, TE | • No specific representation primary and secondary data<br>• Global instead regional CF<br>• Socioeconomic impacts: unaddressed |
| [41] | • Simulation and optimization model<br>• Urban residential area (30 households in India)<br>• PV installations and batteries<br>• FU: 1 kWh annualized energy output<br>• Cradle-to-gate + recycling + transport<br>• IC: EPBT, NER, CEF | • None |
| [7] | • Peer-reviewed scientific publication (2005–2013)<br>• 10 articles | • Mapping-solution: cut-off criteria<br>• Double counting<br>• Integrating life cycle emissions in optimization problem (biases)<br>• Techn. representativness—solution: common measurement points<br>• Multifunctional processes—Solution: avoid system expansion |

**Table A2.** Overview and summary of most relevant studies (CF = characterization factor; FU = functional unit; IC = impact categories FE = freshwater eutrophication; HTP = human toxicity potential; PM = particulate matter; PO = photochemical oxidants; TA = terrestrial acidification; TE = terrestrial ecotoxicity; PV = photovoltaic; EPBT = energy payback time; NER = net energy ratio; CEF = carbon emission factor; LCIA = life cycle impact assessment; LCOE = levelized cost of electricity; GWP = global warming potential; US = United States of America; ESM = energy system model; source: own compilation).

| Reference | Analyzed System | Opportunities & Limitations |
|---|---|---|
| [8] | • LCIA: ReCiPe 2008 (9 IC)<br>• Time frame: 2010–2060 (10a time steps)<br>• Partial equilibrium energy system model with cost minimal combination of resource, conversion and end-use<br>• LCIA indicators: quantification of each resource, conversion and end-use technology, region and time step<br>• Energy scenarios: (i) Modern JAZZ, (ii) Unfinished SYMPHONY, (iii) HARD ROCK<br>• Global multi-regional MARKAL energy system model | • Lack: other industrial and service sector<br>• Missing: future changes of production volumes<br>• Non-commercial energy carriers: out of scope<br>• Differentiation direct and indirect impacts: not always straightforward<br>• Infrastructure contribution/ decommissioning: temporal inaccuracies<br>• Direct, indirect and infrastructure: redefined to match region |
| [42] | • Simulation and optimization model<br>• Tucson, Lubbock and Dickinson, US<br>• Assets: PV installations, wind turbines, batteries, biodiesel generator, fuel cells, electrolyzers, H2 tanks<br>• FU: 1 kWh generated electricity<br>• Cradle-to-gate<br>• IC: LCOE and GWP | • Limit: Utility-scale study incorporating the social cost of carbon<br>• Extend study towards a sustainability assessment |
| [9] | Issues of combining LCA and ESM<br>• Double counting<br>• Import, export and emission targets<br>• Spatial differentiation<br>• Temporal differentiation<br>• Biomass emissions<br>• Multi-functional processes<br>• Future performance of technologies | • Lack: feedback to optimization results<br>• Missing technological details for industry data, Improved LCA/ESM integration by:<br>• Standardization<br>• Centralized database<br>• Criteria for matching technologies<br>• Qualitative aspects (society, politics, risks) |

## Appendix B. Life Cycle Inventory Data

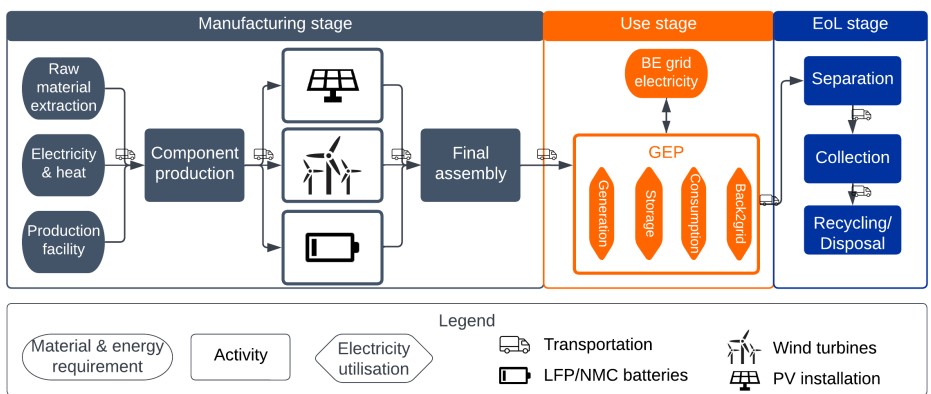

**Figure A1.** System boundaries for evaluation of the RPZ (BE = Belgium; RPZ = Research Park Zellik; Back2grid = electricity fed back to the grid; EoL = end-of-life; LFP = lithium iron phosphate; NMC = lithium nickel manganese cobalt oxide; PV = photovoltaic, source: own compilation).

### *Appendix B.1. Manufacturing Stage*

Specific LCIs of PV installations and lithium batteries are presented by Huber et al. (2023) [39]. As silicon modules are found to be the dominating technology globally, a 3 kWp single-Si PV panel is chosen for this study [39,43]. The selected dataset covers the single-Si panel, electric installation, a 2.5 kW inverter being replaced after 12.5 years, the mounting structure and the electricity required to mount the installation on the roof [17]. For the wind turbines, the dataset of onshore wind turbines with an installed capacity of 2 MW is chosen in alignment with the DOF. The selected dataset for electricity production from onshore wind includes a 2 MW onshore wind turbine, composed of the rotor blades, the rotor hub, the extender, the nacelle, electronics, a steel tower, and the foundation, lubricating oil and its treatment, the corresponding network and transportation. The reference technology is the V80/2 MW wind turbine manufactured by Vestas [17]. Besides the two dispatchable assets in the RPZ, there are two LIBs installed: (i) one LFP and (ii) one NMC. Therefore, these two LIBs are included in this study and specific LCIs for stationary LFP and NMC batteries are obtained from Le Varlet et al. (2020) [22]. Both LCIs of the LIBs contain the battery cells, a module casing, a battery management system and the respective manufacturing infrastructure, transportation and energy impacts.

### *Appendix B.2. Use Stage*

Equally important to the manufacturing impacts is the operational lifetime of the assets as they diverge (see Table A3). Thus, some assets might be required to be replaced earlier or later than others. However, for some assets, a replacement can be an essential driver for the environmental impacts and costs. As DOF simulates only one year, no asset degradation or replacement is included. The lifetime is a crucial parameter, as it is used to calculate the generated or stored electricity of each asset. Afterwards, the manufacturing impact of the dispatchable assets, such as PV and wind installations, can be divided over the generated electricity, representing an impact per kWh. In the case of stationary battery storage, only the discharged electricity is called from DOF, including the battery's efficiency and potential resulting losses. Besides the electricity generation and storage, only lubrication for the rotor operation of the wind turbines is considered. Other maintenance impacts, such as water for cleaning the PV installations, which have a negligible impact on the total environmental impacts, are neglected [43,44].

**Table A3.** Calendric lifetime assumptions of different assets (PV = photovoltaic; LFP = lithium iron phosphate; NMC = lithium nickel manganese cobalt oxide).

| Assets | Lifetime (Years) | Source |
|---|---|---|
| PV installation | 25 | [13] |
| Wind turbine | 25 | [13] |
| LFP battery | 19 | [22] |
| NMC battery | 18 | [22] |
| Energy system | 25 | Own assumption |

*Appendix B.3. End-of-Life Stage*

Due to their novelty and long lifetime, waste streams of renewable energy carriers and batteries are currently still limited, and some treatment facilities, infrastructure, knowledge and data for the EoL treatments are not available. Even though the European Union passed the Waste from Electrical and Electronic Equipment Directive to enforce the recycling of PV installations in Europe, most of the single-Si PV modules are nowadays treated in facilities specialized in recycling laminated glass, metals or electronic wastes [45,46]. Thereby, the primary materials, such as glass, aluminum and copper, can be recycled, while the remaining parts, such as the cells or other plastic components, are incinerated. At the same time, these treatment options allow recovery of materials, in the case of single-Si PV installations, recovery rates for glass vary from 59 to 75% and for nonferrous metals between 13.5 and 21.8% [47]. To include the recycling of PV installations, their LCI is obtained from Frischknecht et al. (2020) [21]. The EoL treatment of wind turbines is not less complex than for PV installations. After collection, various treatment options for different components, such as recycling, incineration, component reuse or landfill, exist. The ecoinvent dataset for electricity from wind turbines takes into account recovered materials of the large components and gives credits for steel, iron, copper, aluminum, glass, plastics and concrete [29,36]. A similar modeling approach is chosen for the batteries; at their EoL, both batteries are collected and treated. To improve recovery efficiency, the battery cells are treated in a pyrometallurgical recycling process, accompanied by hydrometallurgical treatment of the slag, which allows recovery of 95% of copper, iron and nickel, and 70% of lithium [23,37]. The aluminum and steel used to manufacture the battery casing are assumed to be fully recovered, while the battery management system is classified as electronic and the remaining plastic as plastic waste [23]. Similar to LCIs for the manufacturing of PV installations and batteries, inventories for EoL of the same assets are detailed in Huber et al. (2023) [39]. An overview of the recycling rate for different materials is presented in Table A4. To account for the recycling of PV installations, wind turbines and batteries, recovered materials and additional impacts required in the recycling process, such as energy, are included in the model.

**Table A4.** Recovery rates of various assets [21,23,48]. Included rare earth materials are lithium and cobalt, for which recovery rates of 70 and 95%, respectively, were chosen [19,23].

| Materials | PV Installations | Wind Turbines | Batteries |
|---|---|---|---|
| Metals | 13.5–21.8% | 90% | 95% |
| Rare earth materials | 0% | 0% | 70–95% |
| Plastics | 0% | 0% | 0% |
| Glass (fiber) | 59–75% | 40% | n.a. |
| Concrete | n.a. | 90% | n.a. |
| Gravel | n.a. | 90% | n.a. |

## Appendix C. Additional Results

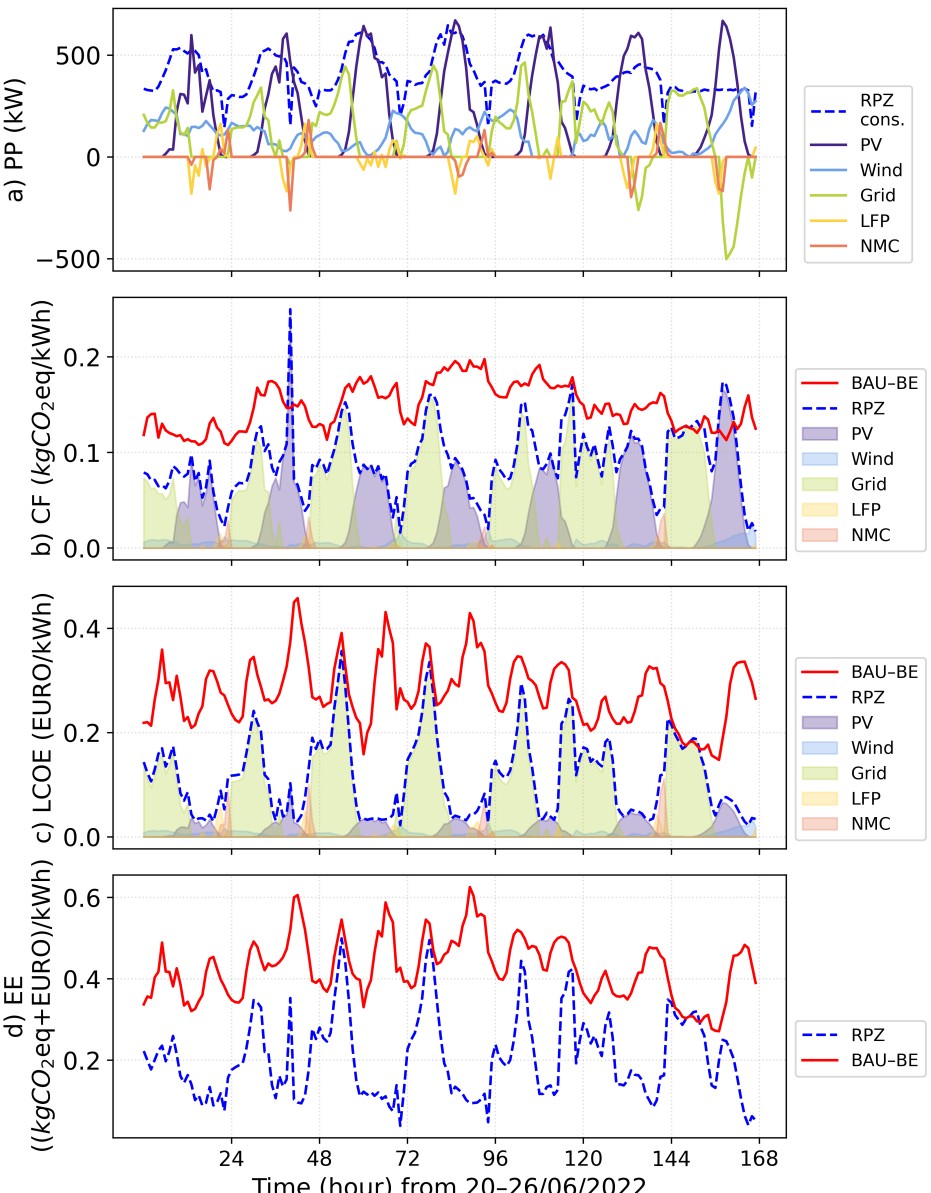

**Figure A2.** Hourly electric power production and consumption by the different assets of the RPZ (positive) and discharged electricity from the stationary batteries and electricity feed back to the grid (negative, subplot (**a**)), the CF (subplot (**b**)), the LCOE (subplot (**c**)) and the EE (subplot (**d**)) of a week in summer in 2022 (PP = power production; CF = carbon footprint; LCOE = levelized cost of electricity; EE = eco-efficiency; BAU = business as usual; RPZ = Research Park Zellik; cons. = consumption; PV = photovoltaic; LFP = lithium iron phosphate battery; NMC = lithium nickel manganese cobalt oxide battery; source: [13] and own compilation).

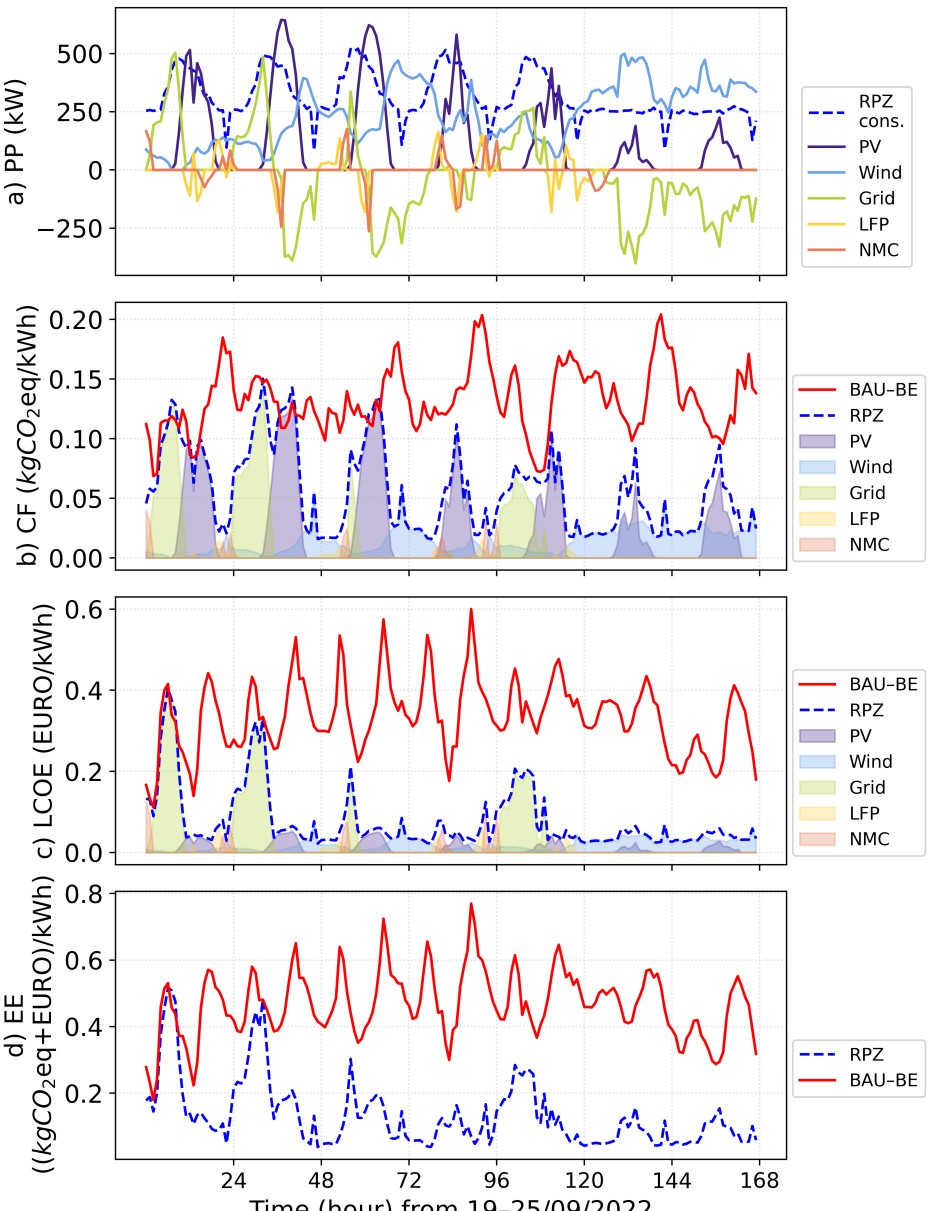

**Figure A3.** Hourly electric power production and consumption by the different assets of the RPZ (positive) and discharged electricity from the stationary batteries and electricity feed back to the grid (negative, subplot (**a**)), the CF (subplot (**b**)), the LCOE (subplot (**c**)) and the EE (subplot (**d**)) of a week in autumn in 2022 (PP = power production; CF = carbon footprint; LCOE = levelized cost of electricity; EE = eco-efficiency; BAU = business as usual; RPZ = Research Park Zellik; cons. = consumption; PV = photovoltaic; LFP = lithium iron phosphate battery; NMC = lithium nickel manganese cobalt oxide battery; source: [13] and own compilation).

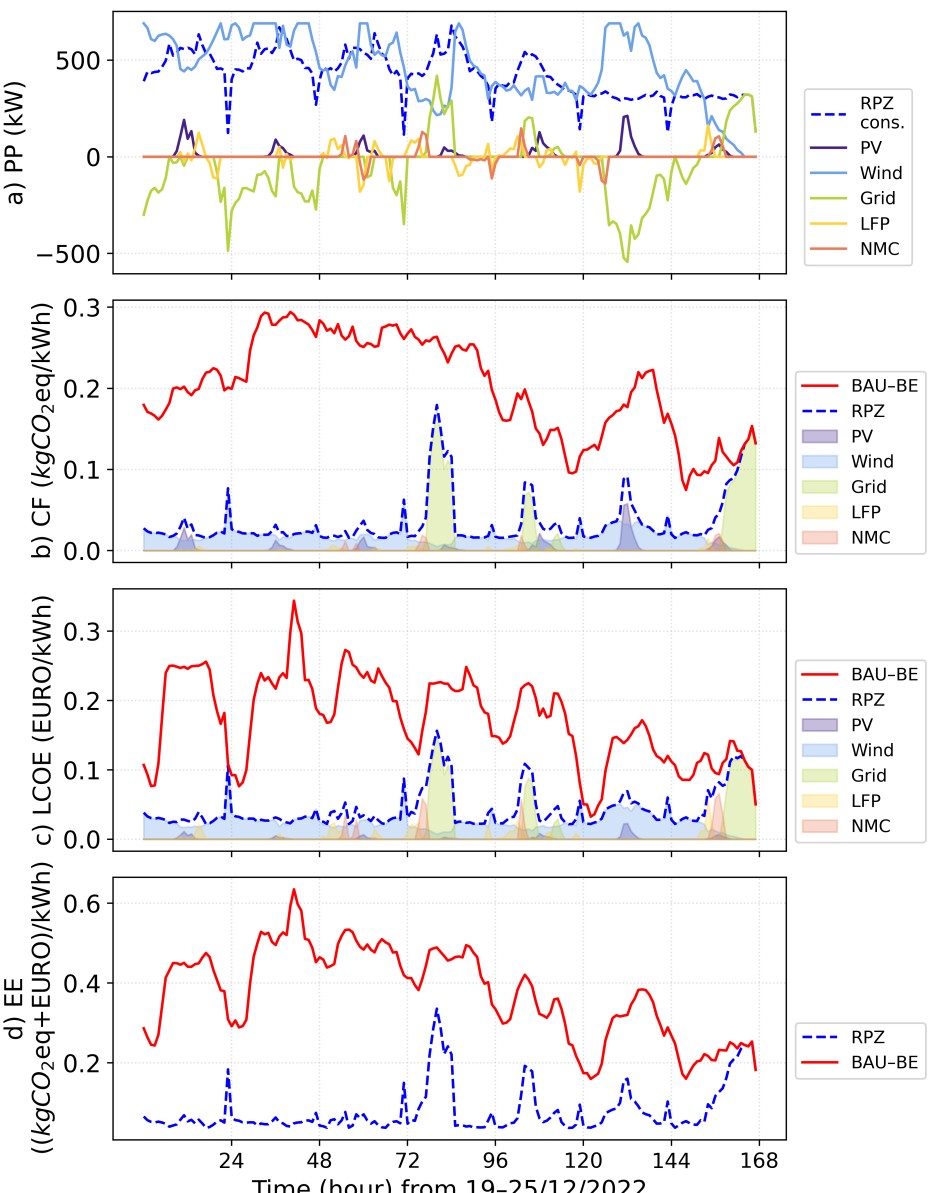

**Figure A4.** Hourly electric power production and consumption by the different assets of the RPZ (positive) and discharged electricity from the stationary batteries and electricity feed back to the grid (negative, subplot (**a**)), the CF (subplot (**b**)), the LCOE (subplot (**c**)) and the EE (subplot (**d**)) of a week in winter in 2022 (PP = power production; CF = carbon footprint; LCOE = levelized cost of electricity; EE = eco-efficiency; BAU = business as usual; RPZ = Research Park Zellik; cons. = consumption; PV = photovoltaic; LFP = lithium iron phosphate battery; NMC = lithium nickel manganese cobalt oxide battery; source: [13] and own compilation).

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
