# Peer review of "Eco-Efficiency as a Decision Support Tool to Compare Renewable Energy Systems"

_energies, doi:10.3390/en16114478_

Round 1
Reviewer 1 Report
I read the article "TiEco-efficiency as a decision support tool to compare renewable energy systems" with interest and I have a comment concerning the main indicator used in the article, the eco-efficiency : It is not clear to me how is calculated the eco-efficiency indicator presented in equation (2) and especially the reason why the eco-efficiency is dimensionless.
I think also that the authors should explain in more detail how the sum between a value measured in [kgCO2eq/kWh] and a value measured in [EURO/kWh] is dimensionless.
Author Response
The authors would like to thank the reviewer for the comment. We agree, that the unit is not dimensionless, but should be (($kgCO2eq$ + EURO)/kWh). Changes have been made throughout the manuscript.
Furthermore, in subsection 2.4 Eco-efficiency the following phrases were added to provide further clarification: “Thereby, the suggested EE does not represent the original definition of efficiency, which is input over output. Instead, the propose of the EE is to provide a single metric to compare different energy systems, computed of the CF and the LCOE. This approach is not new anymore and is applied in Suh, Lee and Ha (2005) [10,30]. Heijungs (2022) describes this procedure particularly relevant for customer or clients, that aim on minimizing both their environmental impact and their costs at the same time [10]. Therefore, the EE is computed for each hour over one year by applying Equation (2):” (line 188-194).
Reviewer 2 Report
The article discusses a study on the hourly, marginal, and seasonal impacts of a decentralized energy system on eco-efficiency (EE). The study focuses on the Research Park Zellik (RPZ), located in the Brussels metropolitan area, and assesses the hourly EE for 2022 using a cradle-to-grave life cycle assessment (LCA) and a levelized cost of electricity (LCOE) calculation. The study applies an existing design optimization framework and analyzes consumption data from smart meters. The article demonstrates the potential benefits of a mixed RES energy system and highlights the importance of considering the seasonal variability in energy demand and production.
Parts of the article:
1. The introduction provides a clear overview of the main objective of the study, which is to assess the energy efficiency (EE) of a renewable energy-based residential energy system in Brussels, Belgium. The authors highlight the importance of reducing greenhouse gas emissions and the potential for residential energy systems to contribute to this objective. The introduction also discusses the research gap that the study aims to address.
2. The materials and methods section describes the methodology used to conduct the study, including the data sources, the life cycle assessment (LCA) approach, and the cost-optimization model used to determine the energy mix of the residential energy system. The authors provide a detailed explanation of each step, making it easy for readers to understand the process.
3. At the beginning of the results section authors presents the validation of the model and show that the selected four weeks are a valid representation of other weeks in the same year. Then, the seasonal and geographical impacts of the energy system on electricity production and consumption, interlinked capacity factor (CF) and levelized cost of electricity (LCOE) have been explored. The analysis shows that the RPZ energy system outperforms the business-as-usual (BAU) scenario in terms of CF, LCOE, and energy efficiency during most hours of the spring week.
4. In the discussion section authors discuss the strengths and limitations of the study and suggest areas for further research. The discussion also provides insight into the practical implications of the findings, such as the need to properly size stationary batteries and the benefits of a mix of renewable energy technologies.
5. The conclusion summarizes the main findings and provides recommendations for future research and highlights the potential benefits of renewable energy-based residential energy systems. Overall, the conclusion effectively ties together the different parts of the article and provides a clear summary of the study.
I would accept the article after minor revision:
- Some of the sentences are too long or too convoluted. Please rephrase convoluted sentences to improve clarity.
- Provide more information about VUB-internally developed DOF works, because it is impossible to reproduce or evaluate it. It is not described in [13].
Some of the sentences are too long or too convoluted.
Author Response
The authors would like to thank the reviewer for the comments. The authors improved the following aspects:
- Regarding the long and convoluted sentences, the manuscript was revised and too long and complex sentences were broken down, wherever possible. If the reviewer foresees further need, the authors are eager to rephrase specific sentences which the reviewer sees still challenging to understand.
An example is provided with the original and the modified text:
Original text: “This analysis could contribute to increase the understanding of the electricity consumers at what time their electricity is generated by renewable energy technologies and hence, enabling them to schedule their electricity consumption accordingly. Specifically, this analysis can be of interest to large electricity consumers, such as the production industry or to electric vehicle (EV) owners, to provide them with decision support to limit their CC impact and save electricity costs.”
Modified text (line 34-39): “This analysis could contribute to increase the understanding of the electricity consumers at what time their electricity is generated by renewable energy technologies. Hence, the electricity consumers would be enabled to schedule their consumption accordingly. Specifically, this analysis can be of interest to large electricity consumers, such as the production industry or to electric vehicle (EV) owners. Thereby, the consumer are provided with a decision support to limit their CC impact and save electricity costs.” - Regarding the VUB-internally developed DOF, the authors would like to refer the reviewer to the sub-sections 3.3 and 3.4 of the cited reference. All applied equations of the model are explained in detail in this cited reference. The authors would also like to point out, that the focus of their study is not on developing a methodology for an economic optimization model, but rather to apply this model and use its output as input for the eco-efficiency compilation. Thus, a detailed explanation including all mathematical formula of the VUB-internally developed DOF is not considered in this work. However, the authors agree that it can be further elaborated. Therefore the description of the framework has been enhanced and the following text in subsection 2.1 was complemented: "For the present paper, the objective function is to minimize the total cost, comprising investment and operation costs. Once the minimisation objective is selected, the problem is defined as a mixed integer linear problem. The methodology of the optimization-problem formulation is presented comprehensively in Felice et al. (2022) and will therefore not be further detailed as part of this study [13]. " (line 127-132)
Reviewer 3 Report
I would like to congratulate the authors for the great paper.
It investigates the seasonal and geographical differences of an energy system of Research Park Zellik (RPZ) in Belgium, based on the hourly eco-efficiency, for four weeks representing each season. Besides the seasonal impact on the eco-efficiency, the performance of the RPZ energy system is also compared to the Belgium electricity grid mix and to two other energy systems located in Poland and Spain to evaluate the spatial difference in the eco-efficiency. The eco-efficiency is determined as the sum of carbon footprint and levelized cost of electricity.
The obtained results are useful for the consumers of the energy system as they can schedule their consumption to the times when the eco-efficiency of the system is the lowest.
The methodological aspect of the paper is also detailly explained, so this paper can serve as a good reference point for future research in this area.
Author Response
The authors would like to thank the reviewer for the congratulations. No changes were made.